# Comparison of Serum and Urine as Sources of miRNA Markers for the Detection of Ovarian Cancer

**DOI:** 10.3390/biomedicines11092508

**Published:** 2023-09-11

**Authors:** Tomas Kupec, Andreas Bleilevens, Birgit Klein, Thomas Hansen, Laila Najjari, Julia Wittenborn, Elmar Stickeler, Jochen Maurer

**Affiliations:** Department of Gynecology and Obstetrics, University Hospital RWTH Aachen, 52074 Aachen, Germany

**Keywords:** miRNA, ovarian cancer, urine, serum, microarray chip

## Abstract

Ovarian cancer is the second most fatal gynecological cancer. Early detection, which could be achieved through widespread screening, has not yet had an impact on mortality. The aim of our pilot study was to investigate the expression of miRNAs analyzed by a human miRNA microarray chip in urine and serum of patients with ovarian cancer. We analyzed three serum and three urine samples from healthy donors and five serum and five urine samples from patients with ovarian cancer taken at first diagnosis, before any treatment. We selected the seven miRNAs with the highest expression fold change in the microarray chip (cancer vs. control) in urine and serum, for validation by qPCR. We were able to validate two of the seven miRNAs in serum. In contrast to these findings, we were able to validate all of the top seven miRNAs identified in urine using qPCR. The top seven miRNAs in urine identified by microarray chip showed significantly greater differences in expression between patients with ovarian cancer and healthy donors compared to serum. Based on our finding, we can suggest that urine as a biomaterial is more suitable than serum for miRNA profiling by microarray chip in the search for new biomarkers in ovarian cancer.

## 1. Introduction

Ovarian cancer is the second most fatal gynecological cancer after breast cancer Koch-Institute, R., 2017 [1]. Early detection, which could be made possible by widespread screening has, so far, no effect on mortality. Most cases have been diagnosed at an advanced stage for decades.

Biopsies of body fluids such as serum, urine and saliva have become of increasing interest in oncology research in recent years [2,3].

The ubiquitous presence [4,5] and stability [6] of miRNAs has led to their use as biomarkers in numerous disease entities, especially for cancer detection [3,7,8,9]. Circulating miRNAs are an integral part of this ongoing effort to identify potential early markers of the disease. MicroRNA (miRNA) is a subtype of RNA. It is a family of small, 17–24 short non-coding RNAs that post-transcriptionally regulate gene expression by silencing genes Cai Y. et al., 2009 [10]. They also play a crucial role in cell differentiation, proliferation, apoptosis and tumorigenesis depending on the developmental stage [10].

In ovarian cancer, miR-194-5p was found to play a role in proliferating and migrating cells [11].

The study of Iorio et al. [12] showed that the miR-200 family of miRNAs was highly expressed in ovarian cancer tissue compared to healthy ovarian tissue. Different miRNA expression patterns were demonstrated depending on the histological subtype of ovarian cancer. Creighton et al. [13] showed that miR-31 prevents cancer cell proliferation by controlling apoptosis and functioning as a tumor suppressor gene.

There have been several articles and systematic reviews written about ovarian cancer and miRNAs in blood-derived biofluids since the diagnostic relevance of miRNAs was first detected in serum samples by Resnick K.E. et al. [14]. Ghafour et al. [15] described miR-1260a as a diagnostic biomarker in blood for ovarian cancer.

In contrast to blood-derived miRNA investigations, there are just a few studies that examined miRNA profiles in urine in ovarian cancer, namely Zhou et al. [16] and Záveský et al. [17], who investigated the urinary expression of 11 different miRNAs in a group of ovarian cancer patients. It was shown that compared to healthy controls, miR-92a and miR-200b were upregulated in cancer patients [17]. Zhou et al. [15] examined urine samples from patients with serous ovarian adenocarcinoma and found that miR-30a-5p was upregulated compared to healthy controls. This finding also correlated with tumor stage and lymph node invasion.

In most of the studies, miRNA candidates are usually selected from the literature and are then validated by qPCR [18].

Currently, a large number of dysregulated miRNAs in ovarian cancer have been identified by high-throughput RNA sequencing (RNA-seq) and microarray technology. The expression profiles of miRNAs are influenced by many clinical, detection and analytical factors in ovarian cancer. RNA-seq and miRNA microarray are the most commonly used techniques for miRNA profiling in ovarian cancer. In addition, quantitative polymerase chain reaction with reverse transcription (RT-qPCR) is usually performed to validate the miRNA profiles.

The aim of this pilot study was to investigate the expression of miRNAs analyzed with a human miRNA microarray chip in urine and serum from patients with ovarian cancer. To our knowledge, there is a lack of evidence comparing miRNA expression via microarray chip in urine and serum in ovarian cancer.

The use of urine, especially, has many advantages and needs to be investigated in more detail. Compared to blood collection, urine is easier and more often quicker to obtain. Urine collection is painless and patients can provide the sample themselves.

## 2. Materials and Methods

In our pilot study, we analyzed three serum and three urine samples from healthy donors and five serum and five urine samples collected at the first diagnosis, prior to any treatment, of patients with ovarian cancer (Table 1). The diagnosis of ovarian cancer was histologically confirmed during further treatment. Sampling, RNA isolation, analysis of samples by quantitative PCR (transcription and quantitative PCR) and statistics were performed as described by Kupec et al., 2022 according to standard operating procedures established in our laboratory [6].

### 2.1. Sampling

Serum and urine samples were collected from eight volunteers by health care professionals at the RWTH University Hospital in Aachen, Germany.

**Serum.** A total of eight serum tubes (Greiner Bio-One, Kremsmünster, Austria) containing 10 mL of whole blood were collected. Each sample was stored until fully clotted. The samples were then centrifuged at 2500× *g* for 10 min. The serum supernatant was carefully pipetted and separated into aliquots. The samples were stored at −80 °C.

**Urine.** The urine samples were collected in sterile urine cups and immediately stored at 4 °C. Within 8 h, the urine samples were centrifuged in sterile 15 mL conical tubes (Perkin Elmer, Waltham, MA, USA) at 1000× *g* for 5 min then filtered (40 µm filter, Corning, New York, NY, USA) and centrifuged for an additional 30 min at 1000× *g* to remove protein content. Each 12–13 mL of supernatant was then stored in a sterile 15 mL conical tube (Perkin Elmer, Waltham, MA, USA) at −80 °C.

### 2.2. RNA Isolation

RNA isolation was performed using the Qiagen miRNeasy Mini Kit (#217004) according to the user’s manual. Aliquots containing 200 µL of serum and 500 µL of urine were used for isolation. After isolation, the isolated RNA was stored at −80 °C until cDNA transcription.

### 2.3. MiRNA Expression Profiling with Microarrays

Total RNA containing low molecular weight RNA was labelled using the FlashTag™ Biotin HSR RNA Labeling Kit (Thermo Fisher Scientific Inc, Waltham, MA, USA) according to the manufacturer’s instructions. Briefly, for each sample, 8 µL total RNA (isolated from a specific volume of urine and serum) was subjected to a tailing reaction for 15 min at 37 °C, followed by ligation of the biotinylated signal molecule to the target RNA sample for 30 min at room temperature and adding of stop solution. Each sample was hybridized to a GeneChip^®^ miRNA 4.0 Array (Thermo Fisher Scientific Inc., Waltham, MA, USA) at 48 °C and 60 rpm for 18 h then washed and stained on Fluidics Station 450 (Fluidics script FS450_0002) and finally scanned on a GeneChip ^®^ 3000 7G Scanner (Thermo Fisher Scientific Inc., Waltham, MA, USA). The expression values were RMA-normalized and summarized with robust multi-array average (RMA) using oligo package [19] under R 4.1.1. Differential expression analysis between the sample groups were conducted with limma package. miRNAs expressed with a fold change >1.5 and *p*-value < 0.05 between the groups were identified as significantly regulated.

### 2.4. Quantitative PCR Analysis of Samples

**Transcription**. miRNA samples were transcribed and amplified using the multistep TaqMan Advanced cDNA Synthesis Kit (Thermo Fisher, Waltham, MA, USA, A28007). Steps were performed in accordance with the manufacturer’s instructions. As an exogenous control, the synthetic miRNA ath-miR-159a (Thermo Fisher, Waltham, MA, USA, 478411_mir/A25576) was added in a specific amount (6 pM) to the first step (poly-A tailing) of the synthesis kit. After the last step, the samples can be stored at −20 °C until they are used again. Prior to qPCR experiments, samples were diluted 1:10 with 0.1× TE buffer (1× TE pH 8.0 from PanReac AppliChem, Monza, Italy (A2575)) according to the manufacturer’s instructions.

**Quantitative PCR**. Transcribed miRNA samples were analyzed using the TaqMan Advanced miRNA Assay (#A25576 Applied Biosystems, Waltham, MA, USA) combined with the TaqMan Fast Advanced Master Mix (#4444557 Applied Biosystems, Waltham, MA, USA). Detection was performed on the Roche LightCycler 480 Instrument II (#05015243001). Samples and master mix were added to 384-well plates (#04729749001 Light-Cycler 480 Multiwell Plate white, Roche). Samples were diluted with 0.1x TE for correct preparation according to the manufacturer’s instructions. For each miRNA assay and the exogenous control ath-mir-159a, samples were pipetted in triplets (technical replicates) on each plate. We were able to detect the expression of miR-24-3p, miR-23a-3p, let-7c-5p, miR-26a-5p, miR-191-5p, miR-30a-5p, miR-99b-5p in urine with this setup, and the expression of let-7b-5p, let-7c-5p, miR-6126, miR-1228-5p, miR-25-3p, miR-6798-5p, miR-4270 in serum with this setup.

### 2.5. qPCR Statistics

The data from the LightCycler 480 were exported as an MS-EXCEL file and analyzed. The resulting Ct (cycle times) values were analyzed using the ΔCt method. To normalize the data as ΔCt, the microRNA ath-mir-159a was used as a reference gene. The second reference used to calculate miRNA fold change was the difference between samples from ovarian cancer patients and healthy donors. Statistical analysis was performed using GraphPad Prism 10 software. Significant differences were determined using the Student’s *t*-test.

## 3. Results

We investigated the feasibility of using miRNA expression patterns from different bodily liquids in the detection of ovarian cancer. To this end, we isolated total miRNA from urine and serum from three healthy women and five women at the first diagnosis of ovarian carcinoma (Table 1).

Expression levels of 2578 miRNAs were assessed by microarray chip analysis (Figure 1). After normalization of all samples (Figure 1A) we first analyzed the fold change increase or decrease of miRNAs from both fluids comparing cancer patient samples with healthy controls. A cut-off larger than 1.5-fold was considered feasible to generate reliable results. Here, we could already identify miRNAs differently regulated in serum and urine (Figure 1B). We could mostly identify regulated miRNAs in cancer patients with this 1.5-fold threshold (Figure 1C,D), especially in urine where more miRNAs were significantly upregulated in comparison (Figure 1D). Further analysis was only conducted on miRNAs that could be detected with a significance of *p* < 0.05. With these requirements, we identified 98 serum and 87 urine miRNAs that were significantly upregulated in ovarian cancer patients compared to healthy donors.

In order to validate the findings generated by microarray analysis, we chose the seven miRNAs with the highest expression fold changes (cancer vs. control) in urine (Figure 2) and serum (Figure 3), for validation via qPCR. We were able to validate two out of seven miRNAs in serum: let-7b-5p and let-7c-5p (Appendix A). These two miRNAs were identified with highest expression levels in the microarray analysis. In contrast to these findings, we could validate all top seven miRNAs (miR-24-3p, miR-23a-3p, let-7c-5p, miR-26a-5p, miR-191-5p, miR-30a-5p, miR-99b-5p) identified in urine using qPCR (Appendix A).

To illustrate these findings more clearly and to investigate the feasibility of urine versus serum as a source for miRNA marker profiles, we analyzed the difference in expression levels for the two source liquids. We reasoned that a profile with the higher fold change of expression in ovarian cancer patients and healthy controls in general, regardless of the miRNA used, would make a more feasible source for marker detection in the future.

The difference in expression levels between patients with ovarian cancer and healthy controls of the 7 most expressed miRNAs in microarray chip were more pronounced in urine than in serum. In Figure 4 we show the fold change difference of seven miRNAs with the highest expression levels in urine (Figure 4A) and serum (Figure 4B). The difference was significantly higher in urine than in serum (*p* < 0.0006) (Figure 4C). The mean fold change in urine was 5.82 (95% CI: 4.34–7.29) compared to 2.27 (95% CI: 1.10–3.46) in serum. The highest fold change in serum was seen for let-7c-5p (FC = 3.67). The highest fold change in urine was found for miR-24-3p (FC = 7.57) (Figure 4D).

## 4. Discussion

The use of miRNA as a marker in cancer diagnostics in the form of a minimally invasive liquid biopsy has great potential for future cancer care. In our pilot study, we investigated the expression of miRNAs in the most commonly used liquid biopsies, urine and serum, in everyday clinical practice in patients with ovarian cancer using a miRNA microarray chip. The aim of our study was to verify in which medium (urine, serum) the miRNAs show higher expressions by patient with ovarian cancer compared to healthy controls and which medium is more suitable for clinical practice (Figure 5).

Interestingly, the highest expressed miRNAs by microarray chip in our pilot study show greater differences in urine between patients with ovarian cancer and healthy controls. Similarly, the validation rate by qPCR of highly expressed miRNAs, which were detected via microarray chip, is significantly higher in urine than in serum.

Urine is the ideal biofluid for biomarker detection as it can be collected non-invasively and is also routinely collected as part of the diagnosis and treatment of ovarian cancer. Yun et al. [20] validated the stability of miRNAs in urine. Even after seven cycles of freezing and thawing or 72 h of storage at room temperature, urinary miRNA levels remained unchanged.

Some inconsistencies in the detection of miRNAs have also been noted in the literature. The reason could be the relatively small size of the samples, the different nature of the recruited control groups (healthy volunteers or patients with benign diseases), or the different ethnic groups [21]. The parameters during the miRNA analysis for sample preparation, like sample volume, centrifugation duration and speed, can affect the quality of sample but are not standardized [22]. In our laboratory, we have standard operating procedures for miRNA analysis of serum and urine [6]. For example, clot activation in serum samples causes the release of various biological molecules, including miRNAs, from platelets [18]. Hemolysis can also affect the accuracy of the serum probe [23]. In some cases, even the quantifying of hemolysis could be an important step before measuring some miRNAs in serum as a potential marker. This could also be one of the reasons that may have influenced the expression of selected miRNAs in serum in the microarray chip and their subsequent validation.

Experimental approaches used for miRNA isolation are different in studies. The miRNAs could be profiled with microarrays or with next-generation sequencing, followed by validation with qPCR [24]. Just a few published studies on miRNAs are based on microarrays or on next-generation sequencing as a first step. Instead, they usually select miRNA candidates from the literature and then perform validation using qPCR [18]. The initial screening analyses and the criteria for selecting miRNAs for subsequent validation differ in the articles reviewed. Here are four examples to illustrate that notion: Kim et al. [25] investigated the expression level of seven miRNAs in the serum of ovarian cancer patients. The seven candidates had been previously identified in high-throughput profiling studies as the most differentially expressed in ovarian cancer tissues. Elias et al. [26] used an innovative way to analyze miRNA-seq data. Combining small RNA sequencing from 179 serum samples with a neural network analysis produced an miRNA algorithm for diagnosis of ovarian cancer. Kumar et al. [27] performed miRNA profiling by analyzing the methylation status of genomic DNA. The differentially methylated regions of the miRNA gene promoters identified three hypomethylated regions by qPCR in serum and tissue samples. Záveský et al. [28] profiled the expression levels of eight selected miRNAs, comparing tissue, ascites and urine samples. The difficulties offered by urine were well represented in this study. Some miRNAs were downregulated in both ascites and tumor tissue of patients with ovarian cancer, while urine-derived miRNAs were not differentially expressed.

Zhou et al., 2015 [16] identified miR-30a-5p as a potential biomarker for ovarian serous adenocarcinoma in urine using miRNA microarray data. However, the approach used by Zhou et al. 2015 did not include a purification step of the urine, which we provide here in order to isolate miRNA from protein-free or protein-low samples. We have identified problems with comparable techniques as used by Zhou et al. [16] in the past and rarely found strong upregulation of miRNAs when this purification step was missing. This may also be the reason we were able to identify 87 urinary miRNAs that were significantly upregulated in ovarian cancer patients compared to healthy controls in contrast to Zhou et al. [16], who identified only one miRNA that was upregulated in urine in ovarian cancer patients. We could not identify miR-30a-5p as upregulated in serum. Before establishing the new miRNA as a biomarker, further investigations would be necessary to prove this finding, including its examination in blood.

In our pilot study, we used a microarray chip to profile miRNAs in serum and urine, followed by validation by qPCR. In our opinion, there is a lack of data comparing the suitability of urine and serum as biomaterials after using miRNAs based on microarray or next-generation sequencing analysis followed by qPCR validation. We present urine as a more suitable biomaterial for the search of new biomarkers than serum. The most highly expressed miRNAs show higher expression in urine (than serum) in patients with ovarian cancer compared to healthy controls, and the rate of validation of potential new markers by qPCR is also higher in urine than in serum (summarized in Figure 4).

It could be assumed that the suitable, highly expressed miRNAs identified by microarray chip in urine of ovarian cancer patients would not need validation by qPCR. This was only true for the urinary ones. We were able to validate all seven highly expressed miRNAs identified in urine. On the other hand, the highly expressed miRNAs in the serum of ovarian cancer patients need to be validated by qPCR. We could only validate two out of seven selected miRNAs by qPCR. It shows that urine is very sensitive for the search of highly expressed miRNAs by microarray chip in patients with ovarian cancer and delivers solid results that can be validated by alternative methods.

Mutations in the TP53 tumor suppressor gene are found in 96% of high-grade serous ovarian cancers [29]. This gene encodes the tumor suppressor protein p53 and influences the cell cycle by regulating genes responsible for repairing cell damage or apoptosis. The tumor suppressor genes BRCA1 and BRCA2 also play an important role. In patients with high-grade serous ovarian cancer, a BRCA1 or BRCA2 mutation was found in 22% of tumors [30]. Seven urinary miRNAs (miR-24-3p, miR-23a-3p, let-7c-5p, miR-26a-5p, miR-191-5p, miR-30a-5p, miR-99b-5p) and two serum miRNAs (let-7b-5p and let-7c-5p) validated by qPCR of the top seven miRNAs identified by microarray chip showed high statistical significance between healthy donors and ovarian cancer patients, with *p* values < 0.05. We have shown the expression of the target genes of the identified miRNAs in ovarian cancer compared to all cancer types (Table 2). The highest expression associated with ovarian cancer-specific genes shows miRNAs of the let-7 family (let-7b-5p and let-7c-5p) and miR-99b-5p. The median high-expression was found in miR-24-3p. The median expression could be presented in miR-23a-3p, miR-26a-5p, miR-191-5p and miR-30a-5p. There was no association in miR-191-5p with specific genes for ovarian cancer. The only miR-24-3p we detected in our pilot study was associated with the TP53 tumor suppressor gene. We did not find any association between BRCA 1 and BRCA 2 genes and the miRNAs identified and validated in our study.

The limitation of our pilot study is a low number of patients with ovarian cancer. Further studies with more patients are necessary to support the results of our study. Future studies will also investigate the feasibility of using miRNA-sequencing compared to the microarray technique to increase depth of analysis and also detect rare miRNA changes in cancer entities.

The use of urine as a biomaterial, in which circulating miRNA can be detected, offers the great advantage of a gentle, non-invasive procedure. In conclusion, we show in our study that miRNAs as biomarkers in the common body fluids, like urine and serum, could have great potential in cancer research and treatment. We have shown in our study that the highly expressed miRNAs in urine identified using the microarray chip in ovarian cancer showed significantly greater differences in expression between patients with ovarian cancer and healthy controls compared to serum. In addition, the urine miRNAs identified in the microarray chip were more likely to be detected by qPCR. Based on this finding, we can suggest that urine as a biomaterial is more suitable for miRNA profiling by microarray chip in the search for new biomarkers in ovarian cancer than serum.

## Figures and Tables

**Figure 1 biomedicines-11-02508-f001:**
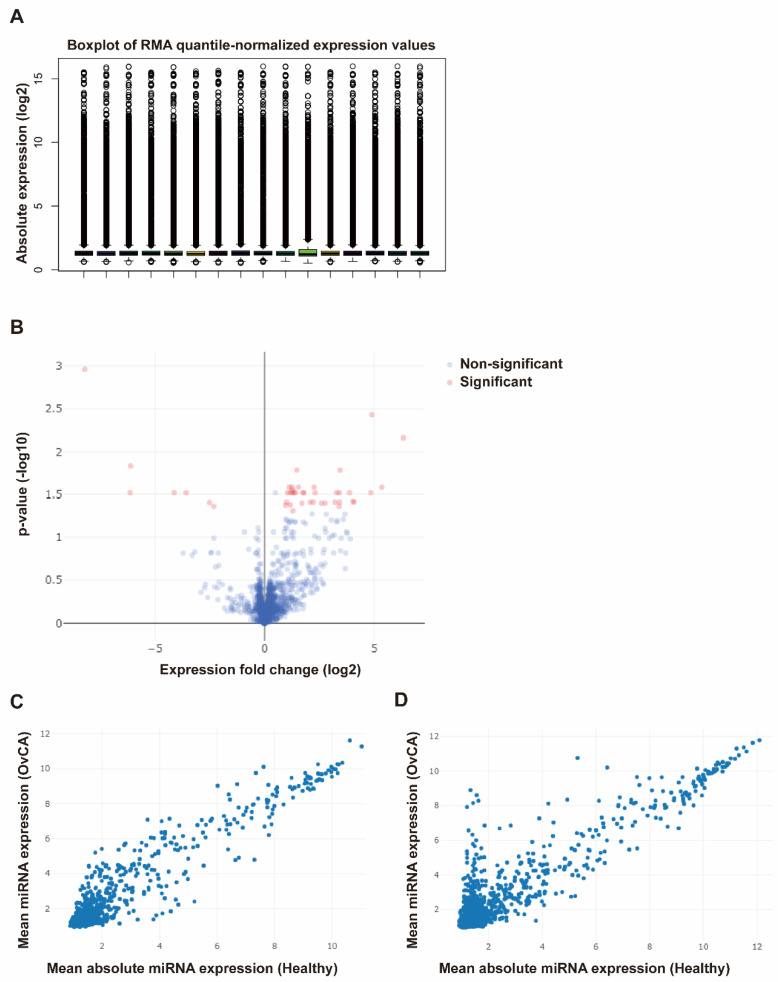
Microarray analysis of miRNA expression from urine and serum samples (**A**) Boxplot of robust multi-array average (RMA)-normalized miRNA expression values (log2). (**B**) Volcano plot depicting significant differences in expression of urine and serum miRNAs. The cut-off values were defined as a *p*-value < 0.05 and a fold change greater than 1.5. (**C**) Comparison of mean miRNA expression between serum samples from ovarian cancer patients and healthy individuals. (**D**) Comparison of mean miRNA expression between urine samples from ovarian cancer patients and healthy individuals.

**Figure 2 biomedicines-11-02508-f002:**
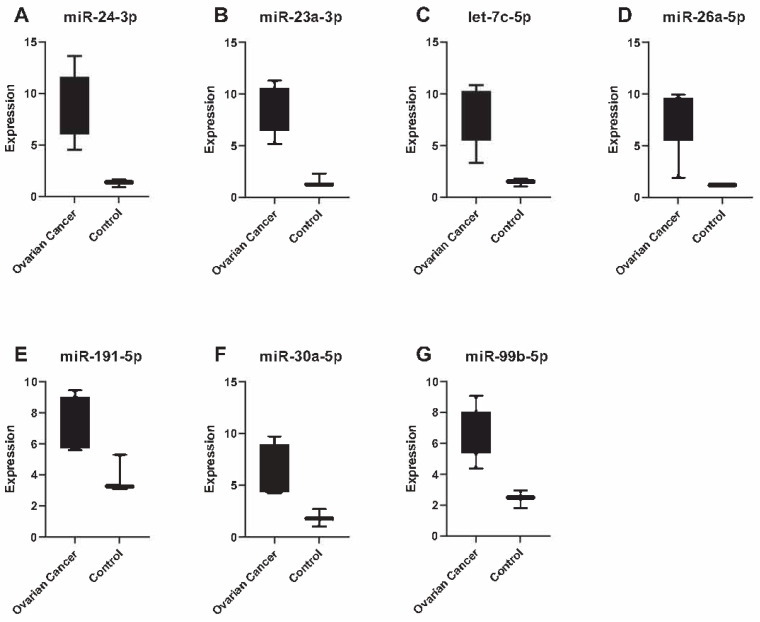
The expression of seven miRNAs with the highest expression levels in urine. The expression is significantly (*p*-value < 0.05) higher in ovarian cancer patients compared to healthy controls. All 7 miRNAs: miR-24-3p (**A**), miR-23a-3p (**B**), let-7c-5p (**C**), miR-26a-5p (**D**), miR-191-5p (**E**), miR-30a-5p (**F**) and miR-99b-5p (**G**) were validated by qPCR.

**Figure 3 biomedicines-11-02508-f003:**
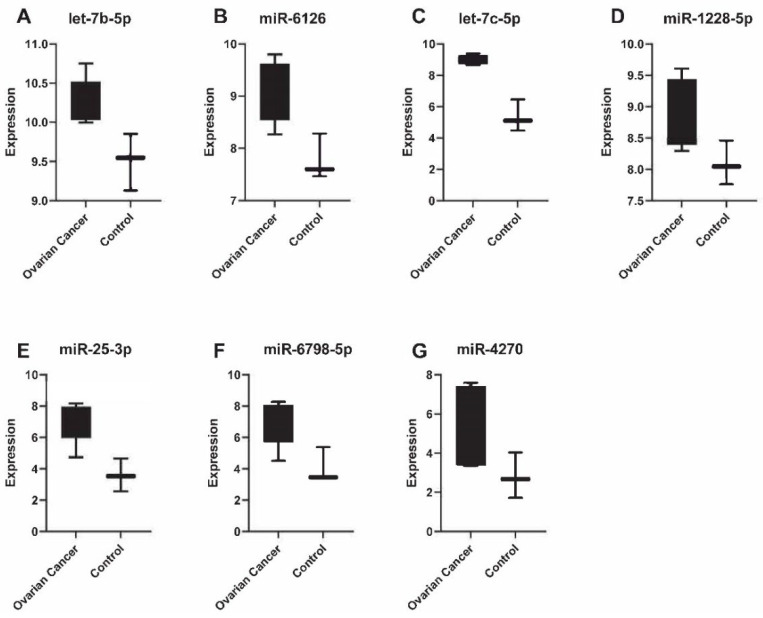
The expression of seven miRNAs with the highest expression levels in serum. The expression is significantly (*p*-value < 0.05) higher in ovarian cancer patients compared to healthy controls. All 7 miRNAs: let-7b-5p (**A**), miR-6126 (**B**), let-7c-5p (**C**), miR-1128-5p (**D**), miR-25-3p (**E**), miR-6798-5p (**F**) and miR-4270 (**G**) were validated by qPCR.

**Figure 4 biomedicines-11-02508-f004:**
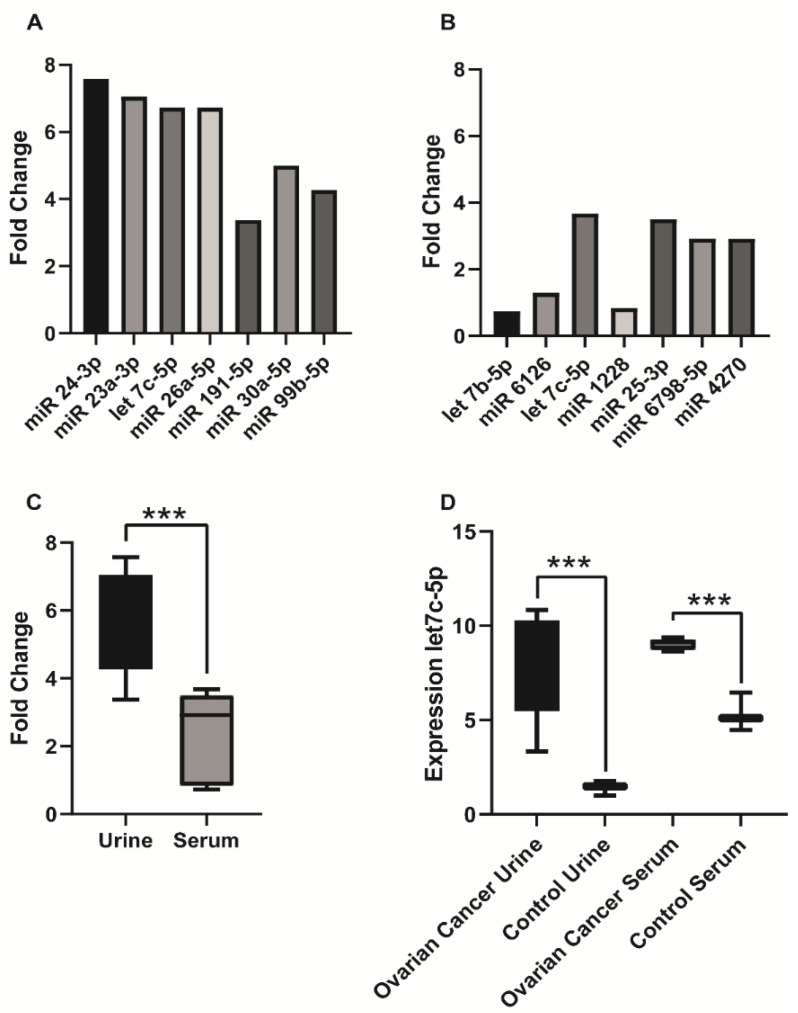
Differences between urinary and serum miRNAs (microarray analysis) (**A**) Fold change difference of seven miRNAs with the highest expression levels in urine (*p* < 0.05). (**B**) Fold change difference of seven miRNAs with the highest expression levels in serum (*p* < 0.05). (**C**) Fold change difference of seven miRNAs with the highest expression levels in serum and urine (*** *p* < 0.001). (**D**) Expression of let-7c-5p in urine and serum (*** *p* < 0.001).

**Figure 5 biomedicines-11-02508-f005:**
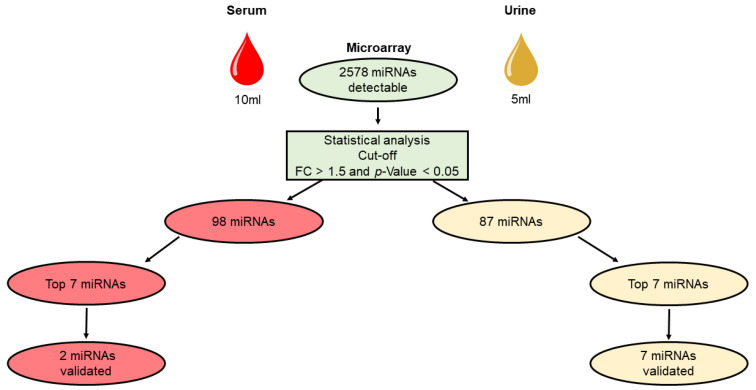
Data processing workflow for identifying highly expressed miRNAs in serum and urine.

**Table 1 biomedicines-11-02508-t001:** Characterization of patients with ovarian cancer.

Sample No.	Age (Years)	Menopausal Status	Histological Subtype	Initial Tumor Stage	FIGO Staging	Nodal Status	Tumor Residues After Surgery
1	57	Postmenopausal	Serous adenocarcinoma	pT3	IIIc	pN1	0 cm
2	67	Postmenopausal	Serous adenocarcinoma	pT3	IIIc	n.a.	>1 cm
3	66	Postmenopausal	Serous adenocarcinoma	pT3	IIIc	n.a.	>1 cm
4	56	Postmenopausal	Serous adenocarcinoma	>pT3	IVb	n.a.	>1 cm
5	55	Postmenopausal	Serous adenocarcinoma	pT3	IIIc	n.a.	>1 cm

**Table 2 biomedicines-11-02508-t002:** Target genes of identified miRNAs in ovarian carcinoma as validated by OncomiR [31]. The table shows miRNAs (column 1); expression value of miRNA in ovarian cancer (column 2); expression category compared to adrenocortical carcinoma, bladder urothelial carcinoma, breast invasive carcinoma, cervical squamous cell carcinoma and endocervical adenocarcinoma, cholangiocarcinoma, colon adenocarcinoma, esophageal carcinoma, head and neck squamous cell carcinoma, kidney chromophobe, kidney renal clear cell carcinoma, kidney renal papillary cell carcinoma, brain lower grade glioma, liver hepatocellular carcinoma, lung adenocarcinoma, lung squamous cell carcinoma, mesothelioma, ovarian serous cystadenocarcinoma, pancreatic adenocarcinoma, pheochromocytoma and paraganglioma, prostate adenocarcinoma, rectal adenocarcinoma, sarcoma, skin cutaneous melanoma, stomach adenocarcinoma, testicular germ cell tumors, thyroid carcinoma, thymoma, uterine corpus endometrial carcinoma, uterine carcinosarcoma, uveal melanoma (column 3); target genes (column 4); and target gene ID (column 5).

	Expression Levels in OvCa (log2)	Expression Compared to All Cancer Types	Ovarian Carcinoma Specific Regulated Genes	Gene ID
hsa-miR-24-3p	10.44	median high expression		
			neuronal differentiation 1	NEUROD1
			poliovirus receptor-related 1	PVRL1
			neurofilament, medium polypeptide	NEFM
			abhydrolase domain containing 2	ABHD2
			tumor protein p53 inducible protein 11	TP53I11
hsa-miR-23a-3p	11.44	median expression		
			fucosyltransferase 4	FUT4
			UDP-GlcNAc:betaGal beta-1,3-N-acetylglucosaminyltransferase 5	B3GNT5
			ryanodine receptor 3	RYR3
			E74-like factor 5	ELF5
			cellular repressor of E1A-stimulated genes 2	ELF5
hsa-let-7c-5p	14.16	highest expression		
			retinol dehydrogenase 10	RDH10
			zinc finger protein 697	ZNF697
			solute carrier organic anion transporter family, member 2A1	SLCO2A1
			syntaxin 17	STX17
			interleukin 13	IL13
hsa-miR-26a-5p	9.65	median expression		
			pleckstrin homology-like domain, family B, member 2	PHLDB2
			UDP-GlcNAc:betaGal beta-1,3-N-acetylglucosaminyltransferase 5	B3GNT5
			hepatocyte growth factor	HGF
			solute carrier family 24	SLC24A4
			homeobox A9	HOXA9
hsa-miR-191-5p	10.57	median expression	no OvCa-specific genes	
hsa-miR-30a-5p	12.28	median expression		
			neuronal differentiation 1	NEUROD1
			UDP-GlcNAc:betaGal beta-1,3-N-acetylglucosaminyltransferase 5	B3GNT5
			galanin receptor 1	GALR1
			neurofilament, medium polypeptide	NEFM
			syntaxin 17	STX17
hsa-miR-99b-5p	18.06	highest expression		
			frizzled class receptor 8	FZD8
hsa-let-7b-5p	16.39	highest expression		
			retinol dehydrogenase 10 (all-trans)	RDH10
			syntaxin 17	STX17
			solute carrier organic anion transporter family, member 2A1	SLCO2A1
			ADAM metallopeptidase with thrombospondin type 1 motif, 15	ADAMTS15
			R-spondin 2	RSPO2

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
