# Peer review of "Comparison of Serum and Urine as Sources of miRNA Markers for the Detection of Ovarian Cancer"

_biomedicines, 2023, doi:10.3390/biomedicines11092508_

Round 1

Reviewer 1 Report

I thank the authors for submitting the article titled " Comparison of serum and urine as sources of miRNA markers for the detection of ovarian cancer" to Biomedicines. The article is generally well-written; however, I do have some recommendations to make prior to publication.

Comments:

1.     The main limitation of this study is the lack of large pool of patient data to support the author's findings that urine more suitable than serum for search of biomarkers in ovarian cancer. More validation is needed to support this finding.

2.     As per author findings “The top 7 miRNAs in urine identified by microarray chip showed significantly greater differences in expression between patients with ovarian cancer and healthy donors compared to serum.”   Studies on miRNA over the last decade suggest that epigenetic alterations may be involved in regulating miRNA expression in various cancers. It would be important and more informative if the authors could add the RNA-seq data and further validate the findings.

3.     I was wondering, have authors thought of using saliva as biomaterial for miRNA profiling ? It would be interesting to test saliva for miRNA profiling by microarray chip and compare the results with the  urine miRNA profiling.

Reviewer 2 Report

The aim of the proposed pilot study was to investigate the expression of miRNAs analysed by a human miRNA microarray chip in urine and serum of patients with ovarian cancer. The authors isolated total miRNA from urine and serum from 3 healthy women and 5 women at the initial diagnosis of ovarian carcinoma, performed a miRNA chip array followed by qPCR validation. Overall, their results demonstrate the advantage of utilizing urine over serum as a biomaterial source for ovarian carcinoma biomarker detection.

The methodology and description of how the authors planned and executed their study is clear and well written. However, I would like to see more information regarding the possible biological relevance of their results, beyond the fact that urine is better than serum and can be utilized for miRNA extraction in ovarian CA patients.

What is the relation between the DF urine miRNA and ovarian serous carcinoma? Were the same miRNAs shown to be DF in ovaries with and without serous CA? Do they regulate some of the genes known to be mutated in OvCA such as TP53 and BRCA1/2? How many OvCA central genes are experimentally proven to be regulated by the study urine DF miRNAs? Do the DF miRNAs regulate pathways that are known to be involved in OvCA, such as PI3K-AKT-mTOR pathway and RAS-MAPK pathway?. Some of the information may be hidden in the blurry Supplemental Figure 3- please expand and elaborate in the manuscript.

Add another figure for the proposed mechanism connecting the DF urine miRNA and OvCA central genes/pathways

The authors mention in the text p<0.05 and in fig 1b they mention adjusted p value. Which one is correct? If adjusted p value is used

Fig 4a- do the bar plots refer to mean fold change? If so, please add SD/SEM.

Fig 2- “All 7 miRNAs could be validated by qPCR”- if qPCR validation of these results was done, please change to All 7 miRNAs were validated by qPCR

Round 2

Reviewer 2 Report

The author addressed my questions. One comment- figure 6 is actually a table- please change its name
